# Consumer Likings of Different Miracle Fruit Products on Different Sour Foods

**DOI:** 10.3390/foods10020406

**Published:** 2021-02-12

**Authors:** Sung Eun Choi, Jeff Garza

**Affiliations:** 1Department of Family, Nutrition, and Exercise Sciences, Queens College, The City University of New York, Flushing, NY 11367, USA; 2Garza Consulting, Grand Rapids, MI 49525, USA; jeff.garza@the-gc.com

**Keywords:** miracle fruit, consumer, acceptability, sensory analysis, taste, sweeteners

## Abstract

Miracle fruit has a high potential as a healthy sweetening enhancer, due to its powerful antioxidant capacity and its unique ability to transform sour taste into sweet taste. The aim of this study was to analyze the effect of different miracle fruit products on the likings of different sour foods. In total, 200 healthy adults (women 55%, 18–65 years old) evaluated five sour foods (apple, goat cheese, lemonade, yogurt, pickle) before and after miracle fruit application. Four commercial miracle fruit products (pills-Y; G; M, powder-P) were randomly assigned to each panelist. The pre- and post-test likings for overall, flavor, texture, and aftertaste were evaluated by using a nine-point scale. The “meeting expectations” was evaluated only in the post-tests. After miracle fruit administration, all the liking scores in yogurt, goat cheese, and apple increased; in contrast, lemonade and pickle liking scores decreased, except lemonade’s texture with the P product. The Tukey post hoc test showed that the pre-to-post increments for overall, flavor, and texture likings in yogurt and in overall and flavor likings in apple using M product were significantly higher than using other products (*p* < 0.05). This study suggests that miracle fruit application can be an effective method for im-proving consumer likings for yogurt, goat cheese, and apple.

## 1. Introduction

There have been considerable research efforts into the development of miracle fruit due to its promising health benefits as a sweetness enhancer, as well as its potent antioxidant capabilities. The fruit of *Synsepalum dulcificum*, also known as the “miracle fruit” or the “miracle berry,” is indigenous to West Africa and is named for its uncommon ability to change a sour taste into a sweet taste [1,2]. Miracle fruit has been developed as a sweetness enhancer. Furthermore, some studies have been performed using miracle fruit to improve food palatability for cancer patients who have been treated with chemotherapy [3,4] and to improve insulin resistance that is induced by fructose-rich chow in rats [5]. In addition to the unique potential of miracle fruit to transform sour food taste into a sweet taste, the anthocyanin and flavonoid compounds of miracle fruit have gained attention because of their use as healthy colorants and flavorants for functional food applications [6].

Although the health benefit of miracle fruit is promising, miracle fruit has some limitations of usage on food because of its inability of modifying purely salty or bitter taste and denaturing ability by heat and high or low pH conditions. More studies are needed to overcome the miracle fruit’s limitations and make possible a practical use of miracle fruit. The sensory properties of miracle fruit are some of the most important factors in regard to consumer acceptability and preferences, especially for foods and drinks. Therefore, it is very important to determine how sensory factors can affect product acceptance and preference [7]. For this reason, the sensory profiles of different miracle fruit products on different food products were examined by trained panelists by using a quantitative descriptive analysis (QDA) [8]. However, the perceptions of a target consumer group could never be represented by a small size of trained panel [9]. As consumers perceive products as a whole from practical perspectives, they usually give different hedonic reactions [10]. Thus, the hedonistic concepts that were perceived by several trained panelists cannot speak for unsophisticated consumers’ diverse perception and should not be considered as a performance indicator of the product in the marketplace [11]. Identifying consumers’ sensory drivers of liking is pivotal to achieve the highest market share [12]. As an example of consumers’ naïve hedonic reaction, it has been reported that consumers may not be willing to consume a functional beverage that contains a detectable unpleasant flavor even with the added health benefits [13].

Despite the auspicious potential health benefits of miracle fruit that have been suggested by previous studies, a concern remains as to whether consumers would accept the unique sweetness enhancing effects of miracle fruit by compromising unmet expectations for traditional flavor qualities of the corresponding foods. Therefore, the objectives of this study were (1) to examine how consumer likings are affected by the effects of different types of miracle fruit products on different types of sour foods, and (2) to explore the drivers of consumer likings of miracle fruit application on different types of sour foods.

## 2. Materials and Methods

### 2.1. Materials

Green apple, goat cheese, lemonade, cucumber pickle, and non-fat plain yogurt were selected as sour food samples in this study due to the following reasons. First, they are popular sour-taste staple foods, each of which having unique texture and flavor; second, they are usually consumed at cold temperatures that do not cause denaturation of the protein miraculin in miracle fruit; last but not least, they may benefit from miracle fruit taste transforming effect as healthy food items.

Granny Smith apple (Robinson Fresh, Eden Prairie, MN., USA), Kirkland^®^ goat cheese (Costco Wholesale Corp., Issaquah, WA., USA), Hubert’s Original^®^ lemonade (Hubert’s Lemonade, Corona, CA., USA), Vlasic^®^ cucumber pickle (Pinnacle Foods Inc., Parsippany, NJ., USA), and plain fat-free yogurt (Stonyfield farm Inc., Londonderry, NH., USA) were used in this study and these food products were bought in local grocery stores in Queens, NY. Four miracle fruit products (three pills and one powder product) that were used in the descriptive analysis study [8] were applied. The information of the miracle fruit products that were used in this study is presented in Table 1, and these miracle fruit products were purchased at Amazon.com. Distilled water (Poland Spring^®^, Poland, ME, USA) was used to rinse the mouth during the taste test.

### 2.2. Participants

A total of 200 healthy volunteers (Women 55%; Whites 23%, Blacks 21%, Hispanics 26%, Asians 24%, Mixed race 6%) aged 18 to 65 years (Mean ± SE = 32.2 ± 0.9, Median = 27 years old) were recruited through the uses of flyers and word of mouth at the Queens College campus in Flushing, New York. The potential subjects were screened for self-reported current illness, food allergies and intolerances, taste disorders, pregnancy, breastfeeding statuses, smoking statuses, and uses of any medications. After receiving detailed information about the experimental procedures, all of the participants provided signed written consent. The protocol was approved by the City University of New York Institutional Review Board under the reference number 2017-0715.

As presented in Table 2, the consumer participants were evenly distributed in most of the demographic characteristics. For sex, race, and household income distribution, an approximately equal division was exhibited in every group. For the age distribution, although the range of ages (18–65-years-old) was wide, the younger age group (18–29 years old, comprising 57% of the total group) primarily comprised our consumer participants. For the education level, the ‘some college’ group was the largest education group, comprising more than half of the total participants.

### 2.3. Procedures

Unlike other consumer taste tests, the consumer taste tests in the present study had three unique components. First, one session was composed of a pre- and post-test for measuring the effects of miracle fruit administration, and the panelists took a break of about 10 min including miracle fruit administration to allow for the miracle fruit products to coat their taste buds. Second, mouth washing or drinking water were not allowed between the samples and were only allowed before each test to prevent the miracle fruit coating from washing out. Third, a random, three-digit number code, which is a standard practice for sensory tests, was not used because the appearances of the served food samples within a set are obviously different. The procedure that was established in the descriptive analysis study [8] was employed in this study.

#### 2.3.1. Sample Preparation

The evaluated sour food samples and their sample sizes were as follows: four pieces of green apple (flesh only, 1 × 1 × 0.3 cm square block/piece in a 2-oz plastic portion cups with lid), goat cheese (one teaspoon in an odorless melanin spoon), lemonade (1 fl. oz. in a 2-oz. clear plastic shot cup), four pieces of cucumber pickle (1/2 oval chip/piece), and plain fat-free yogurt (one teaspoon in an odorless melanin spoon). Toothpicks were used to obtain the apple and pickle samples. All of the prepared food samples for the pre- and post-tests were maintained in a refrigerator (3 ± 1 °C). Ten minutes before each test, the samples were taken out of the refrigerator and set up in an individual plastic tray. The food samples and water were served at 10 ± 1 °C.

#### 2.3.2. Experimental Design

Due to the relatively lengthy procedure required for the pre- and post-test for the evaluation of one miracle fruit product in one session/visit, this consumer test design could not be extended to include the repeated judgements by the same consumer panelists on all four miracle fruit products over multiple visits. As it was also assumed that the effects of the miracle products on consumer perception are large enough to be detected with *n* = 50 per product, each miracle fruit product was assigned to 50 consumers, the rotation was executed in quadruplicate, and all four of the miracle fruit products were evaluated by a total of 200 consumers. Each panelist evaluated the likings for all five sour food samples in one session, with evaluations consisting of pre- and post-tests. The testing order for the five food samples was randomized for each panelist throughout the entire test.

#### 2.3.3. Sample Evaluation Procedure

Each of the 200 consumers performed the consumer liking test in one visit in a temperature-controlled room (20 ± 1 °C) between October and December in 2018. Most of the tests were performed midmorning or midafternoon. The flow chart in Figure 1 summarizes the steps of the experiment procedure for individual panelist.

The consumers were not permitted to eat or drink (except for water) for 1 h before the sessions. The consumers only rinsed their mouths with distilled water immediately before the pre- and post-test, and the drinking of water was not allowed, in order to prevent the miracle fruit coating from being washed out between samples and to simulate real-life eating situations. The pre- and post-evaluations for overall taste, flavor, texture, and aftertaste were evaluated by using a nine-point hedonic scale (1 = dislike extremely, 9 = like extremely) [11]. For “aftertaste pleasantness,” the ‘no aftertaste detected’ answer option was also provided. “Meeting expectations” was only asked in the post-test in relation to the question, “Thinking about this food product that you have had eaten in the past, how well does this sample meet your expectations for the typical corresponding product?” (1 = much worse than expected, 5 = about the same expected, 9 = much better than expected). At the end of each set of scales, an open-ended comment section was included so that consumer panelists can give their individual feedback about the reason of liking/disliking. All five of the food samples were simultaneously presented in a random order.

Just before the start of the pre-test, the panelists were instructed to drink a sip of distilled water that was at the same temperature as the food samples and to wait for at least 30 s. When they were ready, the panelists began to evaluate the first assigned food sample. After all of the evaluations were performed for one sample, and if nothing remained inside of their mouths, the panelists evaluated the next sample. After completing the pre-test, panelists took a 5-min break and proceeded to miracle fruit administration. For the miracle fruit administration, the panelists were instructed to drink a sip of water, wait 30 seconds, and place a miracle pill or powder on their tongue and roll it around very slowly. After the miracle fruit product was melted, the panelists were allowed to start a post-test. In the post-test, the same order of sample testing that was randomly assigned to each panelist in the pre-test was used.

#### 2.3.4. Statistical Analysis

An independent sample *t*-test was conducted to test the effect of sex for the continuous variables and the homogeneity of variances was evaluated by Leven’s test. In the context of one-way ANOVA, all the consumer liking test results were preplanned and thus compared using Tukey’s Honestly Significant Difference (HSD) to test the effect of miracle fruit products, with a significance threshold of *p* < 0.05 for each attribute. The statistical analyses for the liking attributes were carried out using Sen PAQ Version 6 (Qi Statistics, Berkshire, UK) and R 3.5.3 (R Foundation for Statistical Computing, Vienna, Austria). The statistical analyses for other characteristics were conducted using SPSS 25 (IBM Corp., Armonk, NY, USA).

## 3. Results and Discussion

### 3.1. Comparison of Liking Attributes by Sex

The result of independent *t*-test for the liking attribute pre-to-post difference means for each food sample by sex shows significant differences only in the overall (Women = 0.17, Men = 0.86, *p* = 0.01), flavor (W = 0.21, M = 1.00, *p* = 0.006), and texture likings (W = −0.05, M = 0.60, *p* = 0.007) for apple, and in the expectation for goat cheese (W = 6.09, M = 6.74, *p* = 0.042). Men had significantly larger increases in liking for those attributes after miracle fruit applications than women did. For texture liking of apple, women exhibited a decrease. Tilgner and Baryko-Pilielna reported that women can detect sweet and salty tastes better than men, but are worse at tasting sourness [14]. In the survey that was conducted globally by the National Geographic Society, Gilbert and Wysocki observed that women perceived aromas more keenly than men [15]. Given these previous study results, it appears likely that the relatively higher sensitivity of women for sweet tastes and aromas made them more sensitive to the flavor or texture changes that were caused by miracle fruit application, thus allowing them to respond conservatively.

### 3.2. Comparison of Food Samples

The differences in liking data between pre- and post-miracle fruit application that show the miracle fruit’s taste transforming effect most effectively are presented in Figure 2. All the liking values at the pre- and post-tests are reported in Table A1. After miracle fruit products were applied, all of the liking scores for overall, flavor, texture, and aftertaste increased for yogurt, apple, and goat cheese, except for lemonade and pickle. All of the post likings for lemonade and pickles decreased, while only the texture liking for lemonade with the P product increased, which is indicated by a positive value in Figure 2.

Overall, lemonade and pickle had a similar degree of liking decreases after using the miracle fruit products, but a slight difference could be observed between the results of the two food products (Figure 2). For both products, overall liking and flavor liking results exhibited similar patterns and similar degrees of decrease. Thus, it can be assumed that flavor had a direct impact on overall liking. However, it is interesting that the two products exhibited opposite texture liking results for the powder miracle fruit product, with the effect being that lemonade exhibited a liking increase, whereas pickle exhibited a liking decrease. As shown in the consumer panel’s comments in the present study, they did not dislike the additional powdery texture in lemonade; rather, they provided favorable feedback on the texture liking of lemonade because consumers may have been familiar with a powdered lemonade mix product. However, for pickles, consumers commented that the powdery texture of the miracle fruit-treated pickle bothered them because powdery texture in pickles is unexpected and undesirable.

Among the three products that exhibited liking increases (Figure 2), yogurt exhibited the largest increase (difference ranging from 1.54–4.26) for all of the miracle fruit product in all of the likings, followed by goat cheese (0.40–2.17) and apple (−0.08–1.26). It can be interpreted that dairy products seemed to be the product that conveyed the transforming effect of miracle fruit most effectively. Additionally, another interesting result was the small difference that was observed between yogurt and goat cheese. Whereas yogurt exhibited a similar liking increase across the liking categories, goat cheese exhibited the largest increase in aftertaste liking among the four liking categories (overall taste, flavor, texture, and aftertaste). This result indicated the possibility of the use of miracle fruit for masking or improving the aftertaste of goat cheese. For apple, the likings for the texture and aftertaste likings from the P product decreased after miracle fruit product application. It can be assumed that the liking decreases for apple (in terms of texture and aftertaste) may be due to a similar texture effect that explained the disliking for the lemonade and pickle samples. From the QDA study [8], using the same samples as the present study, increases in mouth coating and decreases in crispness were observed in all of the apple samples. Thus, the unexpected additional powdery texture and aftertaste may interfere with the crispness experience of consumers that is valued by consumers.

### 3.3. Comparison of Miracle Fruit Products

Among the five food samples that were tested, lemonade and pickle exhibited liking decreases after miracle fruit application. However, the degrees of liking decrease in lemonade and pickles differed across the miracle fruit products. For lemonade, the M product exhibited the most significant decrease in overall taste and flavor likings, whereas the G product exhibited the most significant decrease in texture and aftertaste likings (Figure 2c). When considering all of the liking change results for lemonade, the P product was most liked because it exhibited the least decreases for pre- and post-testing for all of the likings. However, pickles exhibited more divided results for the most or least accepted products. For pickles, the M product exhibited the least decreases (most liked) for overall and texture likings, while the P product showed the least decrease for flavor liking and the Y product for aftertaste liking, respectively. In contrast, the Y product exhibited the most decrease (least liked) in overall and flavor likings, while the P product and the M product showed the most decrease in texture and aftertaste likings, respectively (Figure 2d).

For yogurt, the M product showed the most liking increases for all of the liking attributes except for aftertaste liking, in which case the P product was followed by the M product (Figure 2e). For goat cheese, the M product showed the most liking increases for all of the attributes except for aftertaste liking, where only the G product was better. For goat cheese, the difference in aftertaste liking with the use of the P product was significantly lower than with the use of the G product (Figure 2b). For apple, the M product was most liked for all of the attributes with no exceptions, though the P product only exhibited liking decreases for texture and aftertaste acceptances (Figure 2a).

These results indicate that the hedonic responses of different miracle products vary depending on the type of food and hedonic attributes.

### 3.4. Potential Drivers of Likings for Miracle Fruit Application

The QDA study using the same samples and procedure as the present study demonstrated that increased sweetness and decreased sourness, which were caused by miracle fruit application, resulted in the separation of food samples on the principal component analysis (PCA) loading [8]. In comparison with the separation of the food samples on the liking results in the present study, the separation of the food samples by ‘PC1 (sweetness)’ and ‘PC2 (sourness)’ from the QDA study indicates significant similarities to the liking results in the present study. After the miracle fruit products were applied, lemonade and pickle were separated from the other food samples by demonstrating mainly sweetness increases and sourness decreases in the QDA study. Moreover, they differed from apple, goat cheese, and yogurt by exhibiting decreases in likings, unlike the other food samples, which exhibited increased likings. As discussed in the QDA study, the larger decrease in sourness for lemonade and pickle seems to be the result of changes in the taste interaction among the four basic tastes that were initiated by the taste shift from sourness to sweetness, which was due to the miracle fruit’s sweetness enhancing effect. In the QDA study, lemonade exhibited the largest decrease in sourness and the second largest increase in off-flavor, with pickle exhibiting the largest decrease in saltiness and the largest increase in off-flavor among the food samples. In particular for the off-flavor, pickle and lemonade exhibited significantly higher increases than any of the other food samples (*p* < 0.001). When considering those associations among sweetness, sourness, and off-flavor, the resulting net changes from the taste interactions between sweetness and sourness seemed to result in the off-flavor increases from the QDA study. This effect was confirmed through discussions with the trained panelists in the QDA study [8]. Furthermore, the separation of lemonade and pickle from the other food samples in the off-flavor data from the QDA study was identical with their separation in the likings, as shown in Figure 2. However, the direction of the data occurred in the opposite direction, which indicates that as the off-flavor effects of the lemonade and pickle increased, their likings decreased. Based on what has been observed above, it can be logically assumed that the imbalance of sweetness and sourness and the resulting significant off-flavor increases can be the drivers for the disliking of lemonade and pickle, unlike for the other sour foods.

These separation results imply that the drivers of likings for miracle fruit-treated sour foods can be different, depending on the types of sour foods. The following discussion concerning the taste-preference relationship according to the type of food can affirm this implication. For yogurt, Barnes et al. demonstrated that sweeter yogurt tastes resulted in the increased likings of these fruit-flavored yogurts by consumers [16]. The yogurt liking results in the present study are consistent with Barnes et al.’s results. However, the present study demonstrated that consumers did not like sweetness increases in lemonade and pickle in contrast to sourness decreases in lemonade and pickle. This result can be supported by the fact that optimal sucrose levels for the most optimal pleasantness of a product depends on the perception of the appropriateness of sweetness, based on the results of a study by Moskowitz et al. [17]. In this study, the degree of pleasantness of the product initially increased and obtained a maximum ideal point of approximately 8–10% sucrose, after which the degree of pleasantness decreased. Other studies have also suggested that the relative optimal sweetness is the important driver of liking in sweetened sour beverages [18,19,20]. Additionally, Leksrisompong et al. predicted that for both sweetness and mouth coating, the absences of bitter and metallic tastes function as consumer drivers of liking for lemon-lime carbonated beverage products [18]. This is a suggestion only for carbonated beverages. However, the prediction for mouth coating effect seems to be related to the increased liking of powder miracle fruit product treated lemonade in the present study. The relatedness implies that mouth sensory attributes can be drivers of liking in lemon lime beverages; thus, this effect can be applicable to the powder miracle fruit usage for carbonated beverages.

For cheese, several studies confirmed that flavor was more important driving force of overall liking than texture in a variety of different cheeses [21,22,23]. Gonzalez Viras et al. observed that consumers preferred cheeses with milder attributes [24], and Bord et al. also reported that consumers described ‘sourness and bitterness’ as being flaws that explained their disliking for blue cheese [25]. In the QDA study [8], goat cheese exhibited decreases in sourness and bitterness after miracle fruit product application. Therefore, the increases of likings in overall, flavor, and aftertaste in the present study can be explained by the mildness that was due to the decreases in sourness and bitterness after miracle fruit application. Many studies specifically indicate that although goat milk is considered to be a good medium possessing the potential for successful delivery of probiotics [26,27], goat milk products do not gain consumers’ sensory acceptance due to the unpleasant sensory features [28,29,30,31]. Costa et al. also pointed out the lower liking scores for goat milk products and have reported that several educational strategies of repeated exposure could boost the acceptance of goat milk products [32]. These findings suggest the need to use flavor enhancer for goat milk products and specifically the flavor enhancing potential of miracle fruit in dairy products.

For apple, there were many divergent results from several previous studies. In the study by Canadian researchers, the majority of the consumer participants favored a sweet and floral flavored apple, compared to tart apples [33]. Additionally, Bonany et al. reported that European consumers preferred sweet apples (68% of the consumers) compared to mildly sweet apples (32% of the consumers) [34]. Carbonell et al. also observed that 30% of consumers preferred an acidic, astringent apple [35]. However, in the study by Tomala et al., Polish consumers identified two preferred apple groups: firm, juicy, acidic apples and sweet, ripe apples with moderate firmness [36]. Bowen et al. also observed the existence of two consumer groups: a group that preferred sweet apples and a group that preferred more acidic apples [37]. Given these previous apple preference results and comparing them with the apple liking results in the present study, sweetness and sourness may be the drivers of liking for apple.

As seen in many previous taste-preference studies, as well as the present study, the optimal preference of taste depends on the unique interaction among basic tastes and other flavors within the food’s specific context. Cardello also concluded in his book that the taste-preference relationship is greatly affected by the appropriateness of the taste that is determined in the overall food/taste context [38]. That conclusion explains the different liking results for each sour food and aids us in understanding the different drivers of likings for each sour food.

### 3.5. Applications of Research Findings

The preceding results and discussion suggest that the differing effects of miracle fruit application on the liking of sour foods should be considered for ideal miracle fruit applications. In particular, when considering the recently growing inclination of Americans toward sour tasting food [39], the application of miracle fruit as a sweetness enhancer should be carefully applied, depending on the specific type of sour food being used.

Of the five food samples that were evaluated, the three samples that exhibited liking increases after miracle fruit product administration were plain fat-free yogurt, goat cheese, and green apple. The present study results also confirmed that nonfat plain yogurt exhibited the most significant impact on consumer likings, followed by goat cheese. This result indicates that such sour dairy food can exhibit the most benefits from miracle fruit application. When considering national and industrial efforts and attention in promoting lower levels of fat and sugar in foods, this result demonstrates the significant potential of miracle fruit application to be an effective flavor enhancer for such low-fat or low-sugar dairy products. The positive liking results for apple also indicate that miracle fruit can be applied to increase the pleasantness of eating healthy nutritious fruits that have strong unpleasant flavors. Additionally, this application seems to be more effective for men rather than women based on a sex comparison, discussed in Section 3.1.

However, the results of liking decreases for pickle and lemonade imply the need for a different approach for the application of miracle fruit for lemonade and pickle. As has been previously suggested, due to the fact that commercial lemonade with 6.25% *w*/*v* added sugar was used in the present study for the practical reason of its popularity, it seemed that the heightened strong sweet taste resulted in a large increase of off-flavor and an overall decrease in likings in the lemonade samples. Therefore, in order to increase the liking of the lemonade, unsweetened lemonade should be used for miracle fruit application, and the application can be used for the development of acceptable low sugar/sugar free or low-calorie foods.

Miracle fruit has a bright future in functional food applications not only because of its unique taste modifying ability but also as a good source of color and antioxidant agents. However, to actualize the beneficial function of miracle fruit, the following drawbacks of miracle fruit should be managed. Miracle fruit does not modify purely bitter, salty, or other sweet tastes and is deactivated by heat and high or low pH conditions—below pH 2 and above pH 12—because miraculin, the active principle of taste modification in miracle fruit, is a glycoprotein. Thus, miracle fruit cannot be cooked or used with other food ingredients that have high or low pH conditions. In addition to these drawbacks, as it takes time for miracle fruit to coat taste buds inside the mouth in order to have the taste modifying effect, miracle fruit should be administered before eating food or meal. When taking these practical points into account, the development of commercial healthy low-calorie sour food products with add-on miracle fruit pill/powder pouch such as fat-free yogurt with miracle fruit in a two-sided container or sugar-free lemonade with a separate miracle fruit powder envelope, appears to be promising. Furthermore, we suggest sushi as another application area that can be benefited from the miracle fruit’s flavor enhancing effect. In the study examining consumers’ acceptance for sushi meals among Czech consumers [40], the authors reported that sushi can be considered as a potential way to increase fish consumption, with the main barrier of sushi acceptance is that sushi is served cold. Since sushi is a cold and sour food, miracle fruit can be applied to improve sushi acceptance among consumers who proclaim to not consume this type of cold sour meal.

Despite promising results, this study had several limitations. First, in the present study, the likings were evaluated for the separate tasting of five individual foods. In a real-world situation, the miracle fruit needs to be applied for both a single specific food and an entire meal. Second, this consumer study could not use the experimental design wherein each panelist could evaluate all four of the miracle fruit products due to the complexity of the pre- and post-tasting process and the lack of motivation of the consumer panelists in participating in a multivisit process. However, despite this limitation, this study provides valuable insight in miracle fruit consumption. Last, as all of the miracle fruit samples were commercial products, their purities and exact formulas could not be identified. For this reason, a comparison of the miracle fruit concentration effects was not possible in this study.

## 4. Conclusions

Consumer likings for overall taste, flavor, texture, aftertaste, and flavor expectations were evaluated by using four different types of miracle fruit products for five common sour foods. The results from this study demonstrated that significant liking differences exist among the different types of sour foods and miracle fruit products. Among the five sour food samples that were evaluated in the present study, plain fat-free yogurt, goat cheese, and green apple exhibited liking increases, whereas lemonade and pickle exhibited liking decreases, after the application of miracle fruit. The order of the positive effects was plain fat-free yogurt > goat cheese > green apple. We also concluded that such consumer liking results can be explained by the consumer expectations for the balance of flavors, mainly sweetness and sourness. If the resulting sweetness increases and sourness decreases from the miracle fruit application are within a range of being accepted by consumers, then the liking for the miracle fruit-treated food will increase. However, the acceptable ranges of the net changes of sweetness and sourness will be different depending on the types of food. Future studies should be conducted to optimize miracle fruit application for a meal that consists of several dishes (not solely for a single food) and for specific consumer groups that have specific disease conditions. These study findings will be useful in providing insights for the food industry for the development of functional food and for the widening of consumer choices for food remedies.

## Figures and Tables

**Figure 1 foods-10-00406-f001:**
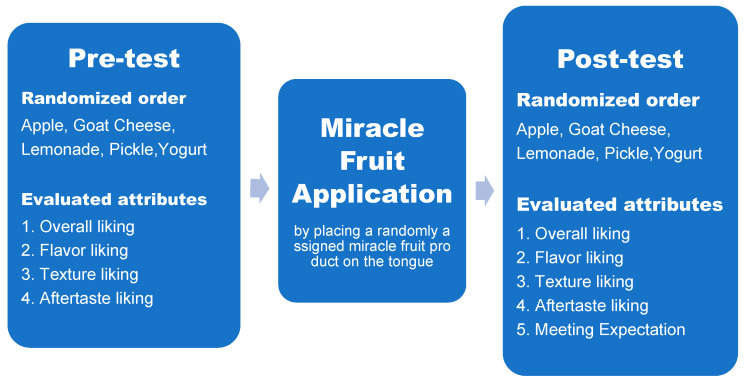
Flowchart of experimental design for individual panelist.

**Figure 2 foods-10-00406-f002:**
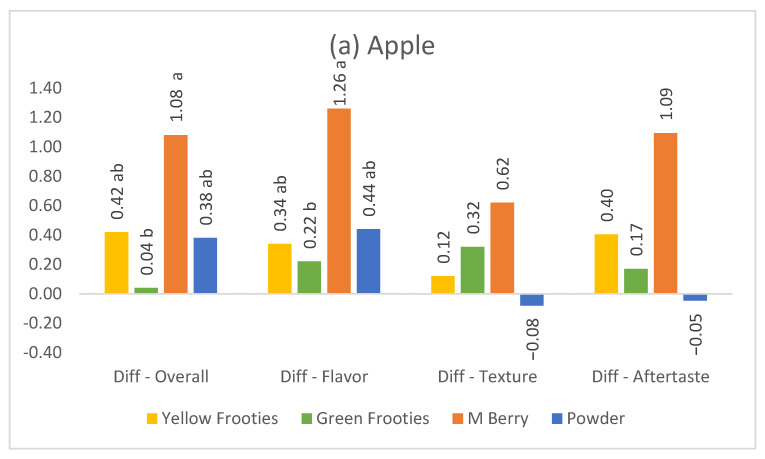
Means of the difference between pre and post-test liking intensity for food samples, (**a**) apple, (**b**) goat cheese, (**c**) lemonade, (**d**) pickle, and (**e**) yogurt on miracle fruit products (Yellow Frooties—Y, Green Frooties—G, M Berry—M, and Powder—P, product details given in Table 1). Means in the same liking attribute labeled with the same letter are not significant different (Tukey HSD test, *p* < 0.05).

**Table 1 foods-10-00406-t001:** Specifications of miracle fruit products used in this study.

Sample Code	Brand	Producer/Distributor	Serving Size	Ingredients
Y	Large Miracle Frooties	Ruby Forest LLC, Wilmington, DE	600 mg (0.21 oz.)/tablet	Dried Miracle Fruit pulp, potato starch, Maltodextrins
G	Miracle Frooties	Ruby Forest LLC, Wilmington, DE	350 mg (0.12 oz.)/tablet	Dried Miracle Fruit pulp, potato starch, Maltodextrins
M	Mberry	Product of Taiwan, MY M FRUIT LLC, Gilbert AZ	400 mg/tablet	Miracle fruit powder, corn starch
P	Sweet Freaks, Miracle Berry Powder	Grow and Packaged in the USA, Bolt Health Supplements, Huntsville, AL	300 mg powder	Miracle berry powder

**Table 2 foods-10-00406-t002:** Demographic characteristics of 200 consumers participating in this miracle fruit product consumer study.

Variables		Frequency
Sex	Women	110
	Men	90
	Total	200
Age (years old)	18–20	31
	21–29	85
	30–49	55
	50–65	29
	Total	200
Race	Non-Hispanic Whites	45
	Non-Hispanic Blacks	41
	Hispanics	53
	Asians	48
	Mixed-race	13
	Total	200
Household Income	<$15,000 a year	38
	$15,000–$29,999 a year	39
	$30,000–$44,999 a year	36
	$45,000–$75,000 a year	38
	>$75,000 a year	49
	Total	200
Education	High school graduate	10
	Some college	108
	College graduate	56
	Post graduate degree	26
	Total	200

## Data Availability

No new data were created or analyzed in this study. Data sharing is not applicable to this article.

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
