# Peer review of "Consumer Likings of Different Miracle Fruit Products on Different Sour Foods"

_foods, 2021, doi:10.3390/foods10020406_

Round 1

Reviewer 1 Report

The manuscript entitled "Consumer likings of different miracle fruit products on different sour foods" shows an interesting study in which a consumer test has been conducted to explore the impact of miracle fruit, or miracle fruit extract, consumption on liking of a set of products. The experimental design is correct and the number of consumers recruited is appropriate for the study. Although the manuscript is well written and results are interesting, some improvements could be made to ease readability and clarify some concepts:
a) Table 2 presents redundant data. It is enough with only one of the columns (absolute frequency or percentage)
b) Please, provide more detail on the statistical analyses used in the study. Why the authors did not used ANOVA test using the gender, age, etc. as factor? Also, if authors studied the different miracle fruits products as one of the factors (table 3), this should be detailed and indicated in the data analysis section.
c) Authors present results in a highly segmented manner. Is the aim of the study to determine if the different gender/age/racial groups respond differently to the different foods/miracle fruit products? If not, I would summarize the results focusing on the specific aim of the study. In addition, some of the groups are not similarly represented (e.g.: age 21-29 [n=85] vs age 50-65 [n=29]) and therefore should not be compared using statistical analyses. Finally, if authors think that some segments are important, should the interaction of some of these factors be studied too (those with similarly represented samples)?
d) In the same manner, tables are too large to facilitate the presentation of the results. I would suggest including only the most relevant, or the significantly differences in data, presenting average and p-value, and not the difference means.
e) Is there any rational behind the samples election (apple, yogurt, ...). If so, please explain it further.

Author Response

We would like to thank reviewers for the careful and constructive comments. Please find our detailed responses to the comments (in red) below. All the corrections that were made using Track Change function in the revised manuscript were specified with line numbers.

Reviewer 2 Report

I have the following suggestions and comments:

  1. The aim written in the abstract should be better defined.
  2. Certain statistical findings should be added to the abstract too.
  3. The following reference should be added, since it is dealing with cold sour food:Đorđević, Đ., & Buchtova, H. (2017). Factors influencing sushi meal as representative of non-traditional meal: Consumption among Czech consumers. Acta Alimentaria46(1), 76-83.
  4. Line 26: there is a technical problem with TAB.
  5. How the homogeneity of the obtained results was evaluated. It should be added to the material and methods part.
  6. It would be good to conduct a Principal component analysis.

Author Response

(The authors gave the same response as above.)
